# Stability of African Swine Fever Virus in Soil and Options to Mitigate the Potential Transmission Risk

**DOI:** 10.3390/pathogens9110977

**Published:** 2020-11-23

**Authors:** Jolene Carlson, Melina Fischer, Laura Zani, Michael Eschbaumer, Walter Fuchs, Thomas Mettenleiter, Martin Beer, Sandra Blome

**Affiliations:** 1Institute for Molecular Virology and Cell Biology, Friedrich-Loeffler-Institut, 17493 Greifswald-Insel Riems, Germany; jolene.carlson@ceva.com (J.C.); Walter.Fuchs@fli.de (W.F.); ThomasC.Mettenleiter@fli.de (T.M.); 2Institute of Diagnostic Virology, Friedrich-Loeffler-Institut, 17493 Greifswald-Insel Riems, Germany; Melina.Fischer@fli.de (M.F.); Michael.Eschbaumer@fli.de (M.E.); Martin.Beer@fli.de (M.B.); 3Institute for International Animal Health/One Health, Friedrich-Loeffler-Institut, 17493 Greifswald-Insel Riems, Germany; Laura.Zani@fli.de

**Keywords:** African swine fever virus, stability, soil, disinfection, risk mitigation

## Abstract

Understanding African swine fever virus (ASFV) transmission is essential for strategies to minimize virus spread during an outbreak. ASFV can survive for extended time periods in animal products, carcasses, and the environment. While the ASFV genome was found in environments around infected farms, data on the virus survival in soil are scarce. We investigated different soil matrices spiked with ASFV-positive blood from infected wild boar to see if ASFV can remain infectious in the soil beneath infected carcasses. As expected, ASFV genome detection was possible over the entire sampling period. Soil pH, structure, and ambient temperature played a role in the stability of infectious ASFV. Infectious ASFV was demonstrated in specimens originating from sterile sand for at least three weeks, from beach sand for up to two weeks, from yard soil for one week, and from swamp soil for three days. The virus was not recovered from two acidic forest soils. All risk mitigation experiments with citric acid or calcium hydroxide resulted in complete inactivation. In conclusion, the stability of infectious ASFV is very low in acidic forest soils but rather high in sandy soils. However, given the high variability, treatment of carcass collection points with disinfectants should be considered.

## 1. Introduction

In the last decade, African swine fever (ASF) has reached an unprecedented geographical spread affecting wild boar and domestic swine in large parts of Europe and Asia, as well as several areas in Africa [1]. The notifiable disease of suids can be accompanied by signs of a viral hemorrhagic fever in domestic pigs and Eurasian wild boar [2]. The virus is transmitted directly between infected swine and wild boar by the oronasal route, as well as indirectly by ingestion of contaminated meat, by fomites or by a contaminated environment. It can also be transmitted by competent vectors, i.e., soft ticks of the genus *Ornithodoros*. They play an important role in Africa, but only in a few areas outside this continent [3]. Despite its limited host range and non-existent zoonotic potential, the socioeconomic impact is high, and many stakeholders are involved [4].

During the first years of the current epizootic that started in Georgia in 2007, infections were mainly seen among pig farms with generally low biosecurity and with incidental spill over to the wild boar population. In the EU, however, the infection survived in the wild boar population independently from outbreaks in domestic pigs [5]. For the transmission among wild boar, carcasses, and the contaminated habitat seem to play a crucial role, together with humans as long-distance spreaders [5].

So far, there are only a few cases in the current European outbreak where the disease was completely eradicated from a country’s wild boar population. One example is the Czech Republic, where the control measures were successfully applied and can be used as a guide for ongoing efforts elsewhere [1]. Another case is that of Belgium, which is very close to becoming free of ASFV as no new wild boar cases have been detected [6]. Both countries followed EU policy to keep the virus concentrated in one zone as much as possible.

An integral part of the control strategy is to search for and remove carcasses as a potential virus source. In this context, the question was raised whether the soil under a removed wild boar carcass should also be removed or treated to prevent virus transmission to other wild boar rooting in the contaminated soil. It was shown that viral genome could be detected in contaminated soil [7], and in order to minimize the transmission risk, simple physical measures such as tilling the soil, but also the application of disinfectants were intensively debated. Commercial disinfectants, as well as lime products, i.e., quicklime and limewater (an aqueous solution of calcium hydroxide), were considered as possible options.

Consequently, our experiments began simply to establish a protocol to isolate ASFV from soil samples but evolved over time as we obtained additional data. We set out to assess the stability of ASFV in soil matrices and to determine how infectivity could be reduced.

## 2. Results

### 2.1. Recovery of ASFV from Yard Soil on Macrophages (Experiment 1)

In this pilot experiment, yard soil was spiked with blood from ASFV-infected wild boar and stored for up to four weeks at 25 °C or 4 °C. A blood-only control was included under the same conditions.

Regarding the blood-only control stored at 4 °C, high variability of the determined virus titers was observed during the first 48 h (Figure 1). Titers of the three biological replicates ranged between 3.75 log_10_ 50% hemadsorbing doses (HAD_50_) per mL and 7.00 log_10_ HAD_50_/mL after an incubation time of six hours at 4 °C. Such a variation did not reoccur at later time points or in the blood-only control stored at 25 °C. In general, virus titers in pure blood decreased clearly after two weeks of storage at either temperature. However, the blood-only controls (4 °C and 25 °C) remained infectious over the entire observation period.

Virus titers in yard soil spiked with infectious blood and stored at 4 °C, or 25 °C generally decreased within the first 72 h (Figure 1). In contaminated yard soil stored at 25 °C, no infectious virus was detectable after 72 h. After one week, however, high variability among the biological replicates was observed in yard soil at both storage temperatures, with virus titers up to 5.50 log_10_ HAD_50_/mL at 25 °C. Hence, we found that ASFV remained infectious in yard soil (pH 6.7) for up to seven days at both temperatures. After two weeks, contaminated yard soil was clearly negative for infectious virus until the end of the study.

Irrespective of the storage temperature, ASFV genome copy numbers were constant over time in the blood-only control and in yard soil samples.

### 2.2. Virus Recovery from Sterile Sand, Beach Sand, Swamp Mud, and Forest Soil on Macrophages (Experiments 2 and 3)

Three different soil types (beach sand, swamp mud, forest soil) were inoculated with blood from an ASFV-infected wild boar and stored at room temperature for up to three weeks. A blood-only control and sterile sand mixed with infectious blood were used as process controls under the same conditions. In these experiments, virus titers in pure blood remained stable over the three-week storage period at room temperature, and no decline in virus titers was observed after two weeks (Figure 2). Virus titers in the sterile sand control, however, decreased constantly over time. Nevertheless, both process controls (blood-only and sterile sand) contained infectious virus over the entire observation period.

In beach sand, high virus titers between 5.50 log_10_ HAD_50_/mL and 6.50 log_10_ HAD_50_/mL were observed directly after application of infectious blood (0 h), but no infectious virus could be detected from three days until the end of the experiment (Figure 2). In contrast, no infectious virus could be recovered from either forest soil specimen (pH 4.1 and 3.2), even immediately after the application of infectious blood. In swamp mud (pH 5.1), however, low residual titers were found directly after the addition of infectious blood. From day three until the end of the observation period, no infectious virus was recovered from swamp mud.

ASFV genome, however, was detectable in all investigated soil types/matrices, and no distinct decline in copy numbers was recorded over the entire observation period.

### 2.3. Virus Recovery from Untreated and Disinfectant-Treated Soil Samples on WSL Cells (Experiment 4)

To evaluate an improved cell culture technique using fluorescent reporter virus and a permanent wild boar lung cell line (WSL) (Figure 3) instead of primary macrophages, beach sand and commercial potting soil were inoculated with blood spiked with WSL-adapted CD2v-deleted ASFV Kenya and stored at room temperature for three weeks. These soil types were chosen because previous experiments showed better stability of infectious virus in sand and more pH-neutral soils.

In addition, the different matrices were treated with two different disinfectants for one or three hours. A blood-only control and sterile sand mixed with infectious ASFV-blood were used as process controls under the same conditions. Cytotoxicity was not observed in cultures after matrix treatment with the respective disinfectants.

In all tested matrices, ASFV genome copy numbers were relatively constant over time (Figure 4). Virus titers in the blood-only and sterile sand controls also remained constant over the entire observation period (Figure 4). Inoculated beach sand and potting soil, however, displayed a steady decline in virus titer over time during the first week. After one week, high variability among the biological replicates was observed in the beach sand. Finally, in both soil types, no infectious virus could be detected after two weeks of storage at room temperature.

Regardless of the matrix (sterile sand, beach sand, potting soil), no infectious virus could be recovered after a one-hour disinfectant treatment (calcium hydroxide or citric acid) at either concentration (Figure 4). ASFV in pure blood was also fully inactivated after treatment with either disinfectant for one hour at room temperature. In disinfectant-treated samples, detectable genome copy numbers were also reduced.

## 3. Discussion

African swine fever is no longer an exotic disease and has established self-sustaining, complex transmission cycles in European wild boar populations. A slow but constant local spread has been observed, as reported by the animal disease notification system [8]. This was rather unexpected as the historical experience did not indicate that wild boar could sustain an endemic infection cycle [9]. Field observations and experimental studies indicate a high lethality [10,11] and low contagiousness, especially in the initial phase of an ASF outbreak among wild boar. The low level of contagiousness requires a rethinking and an adapted approach to control ASF in the wild [12,13]. Evidence suggests that ASF tends to behave more like a long-term (rather stationary) habitat-bound disease (persisting in the ecological niche comprising wild boar, their carcasses, contaminated fomites, and other biotic and abiotic factors) with no tendency to spread rapidly. It is mainly infectious cadavers, combined with the high tenacity of the ASF virus and the low contagiousness, that can maintain the disease within a region [13]. ASFV-contaminated soil rooted by wild boar is one of the habitat factors that could play a role in transmission. Probst et al. [14] reported that wild boar show interest in the soil where carcasses have been found previously, with wildlife cameras documenting animals rooting in the soil even when only bones remained. Furthermore, Estonian colleagues and others have demonstrated viral genome in these soils [7,15].

In our study, we tried to create a data set for a risk assessment of the role of contaminated soil in ASFV transmission and possible mitigation measures.

We demonstrate that virus stability depends on the soil type, pH, organic material percentage, and to a lesser extent, on the ambient temperature. While contaminated sand retains infectivity for weeks, virus stability is very low in acidic forest soils from different locations. This kind of soil is commonly found in northern Germany [16]. Soils are very complex in nature; the interaction of trees, vegetation, animals, microbes, temperatures, location can alter the biology and chemistry of soil ecosystems. In this respect, the existence of different soil types and horizons in forest ecosystems would need further attention. Intermediate times of stability were found in swamp mud and yard soil.

Given the limits of our experimental setup and recognizing that the animal is an even more sensitive detection system, we cannot rule out a persistence of low-level infectivity. While an in vivo bioassay may have had greater sensitivity, we are ethically bound to keep the use of experimental animals to a minimum, and the limit of detection of our hemadsorption test is within the range of what has been shown to be infectious when orally applied to susceptible animals [17,18].

Our results contradict, to a certain extent, previously published studies [19], where water, soil, and leaf litter inactivated ASFV quickly. In their study, Mazur-Panasiuk and Wozniakowski [19] were able to re-isolate ASFV from soil and leaf litter immediately after adding culture supernatant to the matrix, but even a short 3-day incubation caused complete loss of virus infectivity independent of temperature conditions. This is in line with our results from swamp mud, but not from yard soil or sand, where much longer periods of infectivity were observed. Sand, yard soil and potting soil reflect the situation in backyard farm settings and other urban habitats. In contrast, re-isolation immediately after adding the contaminant to forest soil was impossible in our hands. Thus, virus inactivation seems to occur after short contact with the matrix, e.g., due to the acidic conditions in both investigated forest soil specimens (pH 4.1 and 3.2), but it should be mentioned that the ratio between soil weight and infectious blood volume used in our experiments may not always reflect the corresponding ratio found under natural conditions.

Risk mitigation could involve the use of disinfectants despite the obvious limitation that decontamination of soils in fields and forests, which are very different in structure, consistency and composition, is generally difficult and the organic matter in body fluids impairs disinfection [20]. We used citric acid and calcium hydroxide in our study, which both have proven efficiency against ASFV [21,22], the former with known inhibition by blood [23]. It must be noted that in the past, lime products were used in the control of classical swine fever in wild boar, e.g., in Germany. It is assumed that they not only have a disinfectant effect but also repel wild boar. Furthermore, these products were well accepted by the hunters. The application of lime was, therefore, included in the official recommendation of the German government for the use of disinfectants in an epizootic [24]. Despite the above information, it can be questioned whether the application of a basic chemical to acidic soils in the wild boar habitat is appropriate. ASFV is quite reliably inactivated at a pH of below four [25]. Therefore, acidic disinfectants could be more useful, and here, citric acid was our candidate.

In our study, ASFV was inactivated after 1 h of disinfectant treatment. In spiked beach sand and commercial potting soil not treated with disinfectant, ASFV was fully inactivated after two weeks. Untreated blood or sterile sand were infectious for the entire test interval of three weeks with consistent results from virus isolation on macrophages or WSL cells.

ASFV stability is very low in acidic forest soils but rather high in sandy soils. Not all forest soils are the same globally, nor are they homogenous within a single forest. Therefore, given the high variability of wild boar habitats and unpredictable effects of the decay matrix, treatment of carcass locations with disinfectants should be considered when setting up control measures. The powder format of the used chemicals could be beneficial and practical, but regulations on the use of biocides and occupational safety must be considered. Off-label use of commercial products could be an alternative. Disinfectants based on potassium peroxymonosulfate (Trifectant, Virkon S) were recently shown to inactivate ASFV on porous surfaces [26] but had problems with blood under certain circumstances [23]. It is also important to keep in mind that the depth of carcass fluid drainage into different soils may have an impact on disinfection efficacy.

Removal of ASFV-positive carcasses is of utmost importance and remains a critical control measure as the virus may remain infectious in certain soil matrices for weeks. These studies establish useful protocols to isolate ASFV from soil matrices while providing insight into potential management options useful in the field to mitigate transmission.

## 4. Materials and Methods

### 4.1. Collection and Analysis of Soil

Half a kilogram of soil was collected from each of five locations in Mecklenburg-Western Pommerania, Germany (Figure 5). The chosen soil types (yard soil, two kinds of forest soil, swamp mud and beach sand) were based on locations where wild boar are commonly found. In addition, a bag of commercial potting soil (which is similar to yard soil) was purchased to have a more controlled matrix with neutral pH (Appendix A), compared to the acidic forest soil samples. Sterile sea sand was obtained from a lab supplier (Carl Roth, Karlsruhe, Germany). The collected soils were commercially analyzed by an agricultural laboratory in Rostock, Germany (Landwirtschaftliche Untersuchungs- und Forschungsanstalt, LUFA) (Appendix A).

### 4.2. Inocula Prepared for Soil Spiking

In experiments 1–3, whole blood was collected from wild boar experimentally infected with ASFV “Armenia08”. These animal trials were previously conducted for pathogenesis studies [11]. The blood was mixed for 15–20 min with glass beads to remove fibrin. The blood was then stored at −80 °C until use. Since experiments were completed at different time points, stocks for spiking the soil matrices had different titers but differed no more than one log in considered volumes. Infected blood used in experiment 1 had a titer of 7.25 log_10_ 50% hemadsorbing doses (HAD_50_) per mL, blood used for experiment 2 had a titer of 6.00 log_10_ HAD_50_/mL, and blood for experiment 3 had a titer of 7.00 log_10_ HAD_50_/mL.

Experiment 4 used recombinant ASFV-Kenya1033ΔCD2vdsRed virus that was derived from ASFV-Kenya1033 as described by Hübner et al. [27] by substitution of the CD2v ORF (EP402R) from codon 77 to the translational stop codon (386) by a reporter gene cassette. This virus expresses a red fluorescent protein (dsRed) in infected cells and is no longer hemadsorbing due to the deletion of CD2v.

### 4.3. Experiment 1: Recovery of ASFV from Yard Soil on Macrophages, a First Pilot Experiment

In a pilot experiment, 5 g of yard soil were spiked with 400 µL of infectious blood at a titer of 7.25 log_10_ HAD_50_/mL and stored at 4 °C or 25 °C. For comparison, a blood-only control was kept at the same conditions. Blood and soil were tested at time points 0, 3, 6, 24, 48, and 72 h, as well as 1, 2, 3 and 4 weeks. At the respective time points, 5 mL of RPMI-1640 cell culture medium (Thermo Fisher Scientific, Schwerte, Germany) with 10% fetal bovine serum (FBS) and 2% antibiotics (Gibco Penicillin–Streptomycin mix, 10,000 U/mL; Thermo Fisher Scientific) was added to the inoculated soil. Then the soil was agitated in the media by vortexing for 45 s (see Figure 6 for all steps). Next, soil and media were sonicated for 45 s at 4 °C with the settings duty cycle 40%, output 3.5 in a Branson Sonifier 450 (Heinemann Ultraschall- und Labortechnik, Schwäbisch Gmünd, Germany). After sonication, the soil suspension was centrifuged for 30 min at 2500× *g* at 4 °C. The supernatant was poured over a coffee filter, pushed through a 0.45 µm syringe filter (Millex Filter Units; Merck Millipore Ltd., Tullagreen, Ireland) and the filtrate was stored at −80 °C prior to real-time PCR, virus isolation and titration (see Section 4.7).

### 4.4. Experiment 2: Recovery of Infectious Virus from Sterile Sand, Beach Sand, Swamp Mud, and Forest Soil on Macrophages

The collected soils were tested together with two controls: blood-only and blood mixed with 6 g of sterile sea sand (Carl Roth, Karlsruhe, Germany). All soils and controls were spiked with 1.2 mL of ASFV-positive blood with a titer of 6.00 log_10_ HAD_50_/mL and tested in three replicates per condition. The forest soils were extremely dry; therefore, 12 mL of RPMI media with 10% fetal bovine serum were added to all samples for virus isolation. We continued experiment 2 and subsequent experiments at room temperature (25 °C) since we had not seen significant differences between soil stored at 4 °C or 25 °C in experiment 1. We limited our testing to 14 days in experiment 2 as it seemed unlikely that live virus would be detected beyond one week. Sample processing was performed the same way as in experiment 1 (see Figure 6). The resulting filtrates were stored at −80 °C prior to real-time PCR, virus isolation and titration (see below). In addition to the protocol described in experiment 1, we also used a dedicated kit for DNA extraction from soil (DNeasy PowerSoil; Qiagen, Hilden, Germany) from 0.25 g of all matrices and time points.

### 4.5. Experiment 3: Recovery of Infectious Virus from Beach Sand on Macrophages

Experiment 2 was repeated with beach sand and 2 mL of ASFV-positive blood with a titer of 7.25 log_10_ HAD_50_/mL at room temperature, due to inconclusive virus isolation results (data not shown). A larger volume of blood was used to increase the chances for virus detection in this matrix that previously gave mixed results. In the repeated experiment, blood-only and sterile sand were included as controls. Every experimental condition was completed with three replicates.

### 4.6. Experiment 4: Recovery of Recombinant Virus on WSL Cells and Testing of Disinfection Treatments

Virus stock was prepared by mixing 160 mL of supernatant from WSL cells infected with WSL-adapted CD2v-deleted ASFV Kenya virus with 500 mL of defibrinated whole blood from a domestic swine resulting in a final ASFV titer of 6.00 log_10_ TCID_50_/mL.

A 2 mL volume of the spiked blood was used to inoculate 6 g beach sand and commercial potting soil at room temperature. A blood-only tube and sterile sand were again included as controls. The samples were tested at 1 and 3 h, 1 and 5 d, as well as at 1, 2 and 3 weeks, except blood, which was not tested at 1 and 3 h. For sample processing, the same protocol was followed as described above for experiments 1 and 2, with the omission of sonication as the samples were too numerous to sonicate in the 3-h intervals between the first three collections. Every condition was tested in three replicates.

For disinfectant testing, the virus-spiked samples were treated with calcium hydroxide or citric acid at 3.5% or 7.5% each by weight of soil. After the powdered disinfectants were added, the samples were vortexed and incubated for 1 or 3 h at room temperature before further processing.

### 4.7. Virus Isolation and Titration

All soil supernatants were subjected to one blind passage in either primary porcine macrophages (for experiments 1–3) or in a permanent wild boar lung (WSL) cell line (for experiment 4) before virus titration on the same cells. Experiments 1–3 used a hemadsorbing ASFV “Armenia08” virus, whereas experiment 4 used a recombinant ASFV Kenya virus that expresses a red fluorescent protein in infected cells. Due to its CD2v deletion, this virus does not display hemadsorption.

Virus isolation and titration on porcine macrophages (experiments 1–3) were carried out with macrophages derived from peripheral blood mononuclear cells (PBMCs). Blood was collected from healthy domestic pigs in heparin tubes. The whole blood was diluted 1:1 in phosphate-buffered saline (PBS), 35 mL of diluted blood was overlaid on 12 mL of Pancoll (PAN-Biotech, Aidenbach, Germany) and spun at 730× *g* for 40 min at 20 °C with slow acceleration and no brake. The PBMCs were collected and washed twice in PBS and passed over 70 µm nylon strainers to remove any fatty debris. Five mL of concentrated red blood cells were removed from the bottom of the Pancoll preparation, washed once in PBS and subsequently diluted to make a 2% solution for use in the hemadsorption test (see below). The red blood cell solution was stored at 4 °C until use.

PBMCs were cultured in RPMI-1640 medium supplemented with 10% fetal bovine serum (FBS), 2% penicillin and streptomycin (10,000 U/mL) (Gibco, Carlsbad, CA, USA), 75 µL mercaptoethanol (Merck, Darmstadt, Germany) and 2.5 ng/mL granulocyte-macrophage colony-stimulating factor (GM-CSF; Biomol, Hamburg, Germany) for the first day, then with 5 ng/mL GM-CSF from the second day after a media change.

For blind passages, 5.0 × 10^6^ PBMCs per well were seeded into 24-well Primaria plates (Corning, Durham, NC, USA) one day before inoculation. The following day, a volume of 300 µL filtered soil supernatant was inoculated per well with 700 µL of media. One day after inoculation, the macrophages were washed with media once, and the supernatant in each well was replaced with 1 mL of fresh media. Cultures were examined daily for cytotoxicity with a light microscope. The cells were cultivated for 5 days, and then the cells and supernatant were frozen at −80 °C for subsequent virus titration.

Titrations were performed with 7.5 × 10^4^ PBMCs per well in 96-well Primaria tissue culture plates (Corning) seeded one day before inoculation. The following day, the attached macrophages were inoculated with 100 µL of supernatant from the blind passage in ten-fold dilutions from 10^−1^ to 10^−8^. One day after inoculation, 10 µL of a 2% solution of red blood cells were added to each well. The plates were examined daily for the next 4 days with a light microscope. Each well with at least one hemadsorbing macrophage was considered positive. Titers were calculated by the Spearman–Kerber method and expressed as log_10_ 50% hemadsorbing doses (HAD_50_), with a limit of detection of 10^1.75^ per mL.

For virus isolation and titration on wild boar lung cells (experiment 4), WSL cells were cultivated in Iscove’s Modified Dulbecco’s Medium with Ham’s F-12 Nutrient Mix (Thermo Fisher Scientific), 10% FBS and 2% penicillin and streptomycin (10,000 U/mL) (Gibco).

For the blind passage, WSL cells were seeded the day before with 10^6^ cells per well in a 24-well tissue culture plate (Corning). The following day, a volume of 300 µL soil supernatant was inoculated per well with 700 µL of media. The media was changed the next day and replaced with 1 mL of fresh media. The cells were cultivated for 5 days and then frozen at −80 °C for subsequent virus titration.

Titrations were performed with 3 × 10^5^ WSL cells per well in a 96-well tissue culture plate (Corning, Durham, NC, USA) seeded one day before inoculation. The following day, the attached cells were inoculated with 100 µL of supernatant from the blind passage in ten-fold dilutions from 10^−1^ to 10^−8^. For the next 5–6 days, red fluorescence indicative of virus replication was read daily with an epifluorescence microscope. Titers were calculated by the Spearman–Kärber method and expressed as log_10_ 50% tissue culture infectious doses (TCID_50_), with a limit of detection of 10^1.75^ per mL.

### 4.8. Viral Genome Detection via Real-Time PCR

Prior to real-time PCR analysis, nucleic acids from soil samples were extracted using the DNeasy PowerSoil kit (Qiagen, Hilden, Germany) according to the manufacturer’s recommendations. Subsequently, nucleic acids were analyzed using a published real-time PCR assay targeting the ASFV p72 gene [28] in combination with an internal control based on beta actin [29] on a CFX96 real-time cycler (Bio-Rad Laboratories, Hercules, CA, USA). PCR was performed with the QuantiTect Multiplex PCR Kit (Qiagen, Hilden, Germany) in a total volume of 25 µL, with 10 pmol of each ASFV primer, 1.25 pmol of the ASFV probe, 2.5 pmol of each beta actin primer and 2.1 pmol of the beta actin probe. The temperature profile was 15 min initial activation at 95 °C, followed by 40 cycles of denaturation for 60 sec at 94 °C and annealing/extension for 60 s at 60 °C, as recommended by the manufacturer.

Using a dilution series of an ASFV DNA standard, the genome copies in the respective samples were determined to allow harmonization between PCR runs. For the generation of the ASFV standard, DNA from an ASFV “Armenia08” PBMC culture supernatant was extracted using the QIAamp Viral RNA Mini Kit (Qiagen, Hilden, Germany) according to the manufacturer’s recommendations. Subsequently, the DNA concentration was determined by spectrophotometry using a Nanodrop 2000c (Thermo Fisher Scientific), and the corresponding number of genome copies was calculated based on the ASFV genome length using an online tool [30]. No correction was made for any bias potentially introduced by cellular DNA in the culture supernatant.

## Figures and Tables

**Figure 1 pathogens-09-00977-f001:**
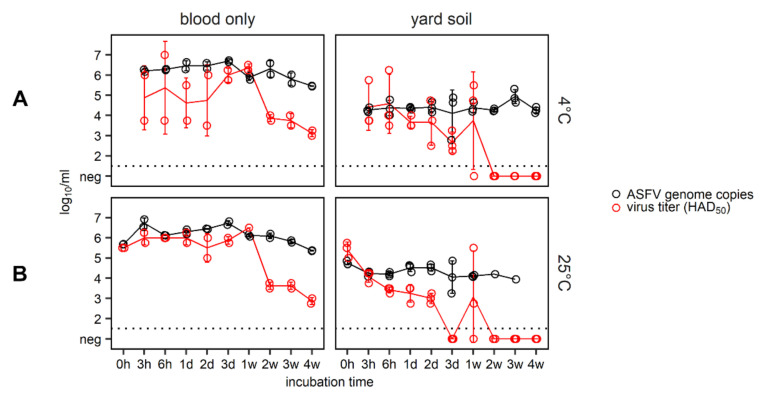
Infectious wild boar blood (7.25 log_10_ 50% hemadsorbing doses (HAD_50_)/mL of African swine fever virus (ASFV) “Armenia08”) and yard soil spiked with 400 μL of this blood were stored at 4 °C (**A**) or 25 °C (**B**). ASFV genome copies per mL are depicted in black and virus titer (as log_10_ HAD_50_/mL) is shown in red. Experiments were completed in triplicates, and each open circle represents an individual replicate. Solid lines and error bars represent the mean and standard deviation. The dotted line is the limit of detection of the virus titration.

**Figure 2 pathogens-09-00977-f002:**
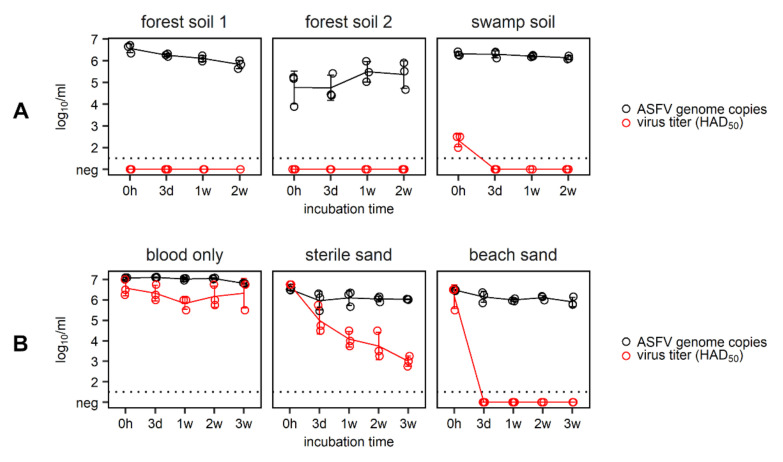
Different soil types spiked with 1.2 mL infectious blood (6.0 log_10_ HAD_50_/mL of ASFV “Armenia08”) and stored at room temperature (**A**). Beach sand and sterile sand were inoculated with 2 mL of infectious blood and also incubated at room temperature (**B**). Blood only samples served as controls in both experiments. ASFV genome copies are depicted in black and virus titers are shown in red. Experiments were completed in triplicates, and each open circle represents an individual replicate. Solid lines and error bars represent the mean and standard deviation. The dotted line is the limit of detection of the virus titration.

**Figure 3 pathogens-09-00977-f003:**
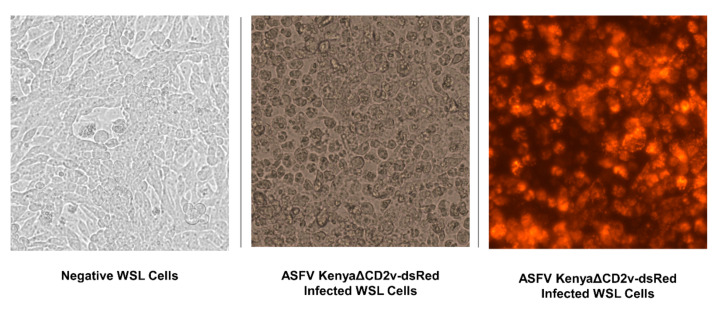
Cytopathic effect and fluorescence in wild boar lung (WSL) cells infected with ASFV KenyaΔCD2v-dsRed.

**Figure 4 pathogens-09-00977-f004:**
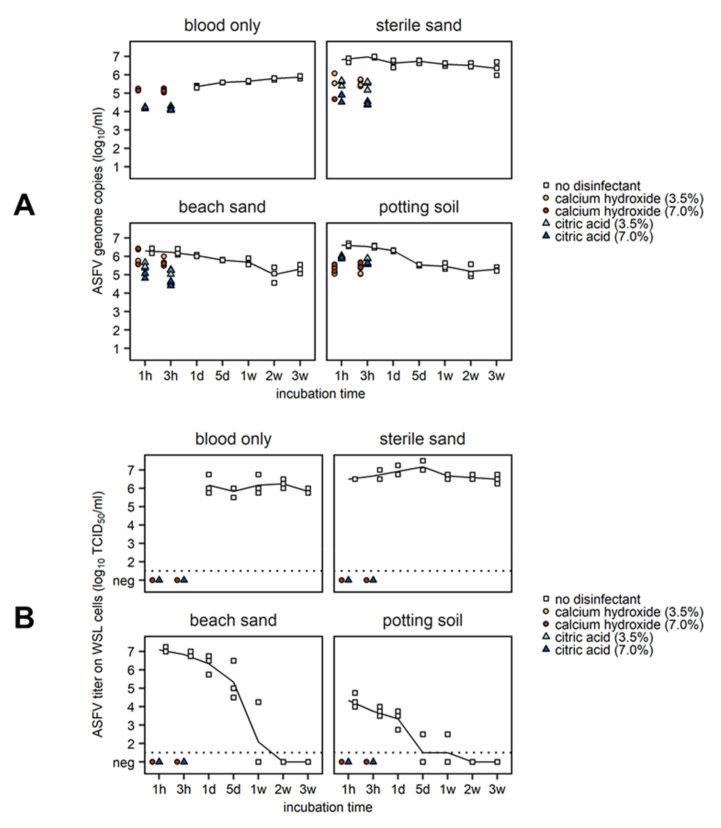
Different soil types were spiked with 2 mL of spiked blood (containing 6.0 log_10_ 50% tissue culture infectious doses (TCID_50_)/mL of ASFV KenyaΔCD2v-dsRed) and stored at room temperature for the indicated times. ASFV genome copies (**A**) and virus titers (**B**) on WSL cells in untreated or disinfectant-treated matrices are shown. Experiments were completed in triplicate. Solid lines represent the mean of the three replicates. The dotted line is the limit of detection of the virus titration.

**Figure 5 pathogens-09-00977-f005:**
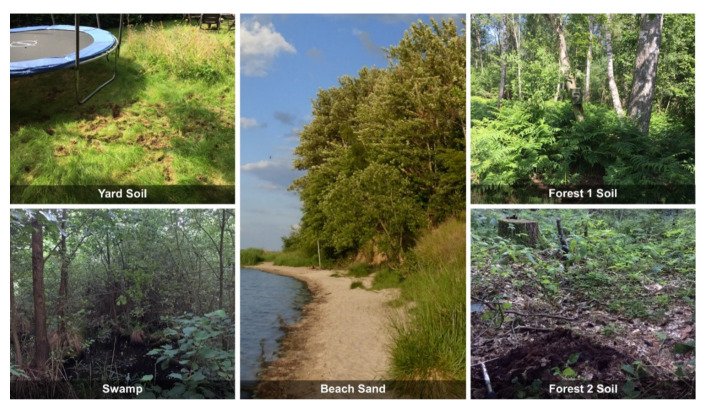
Areas where soil was collected in northern Germany. Sources of yard soil, swamp mud, beach sand, and two forest soils are shown.

**Figure 6 pathogens-09-00977-f006:**
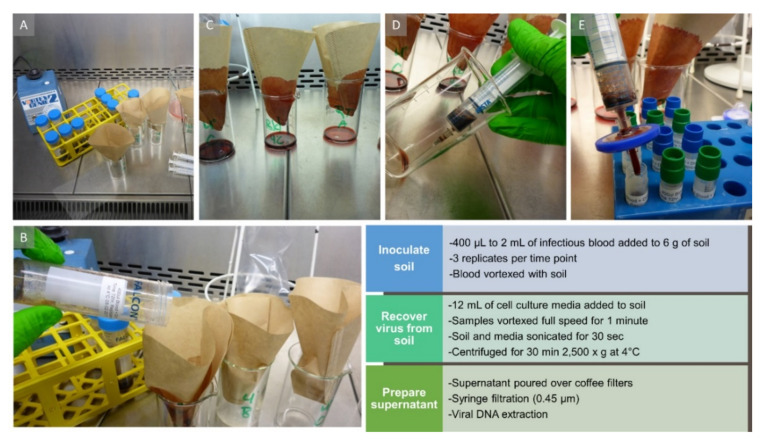
Final downstream protocol for the processing of soil samples for virus isolation and qPCR. Panel (**A**) depicts samples after media and soil matrix were sonicated and centrifuged for 30 min at 2500× *g* at 4 °C. Afterward, the supernatant was poured over coffee filters (**B**). The filtration step is shown in panel (**C**). The filtrate was drawn up with a syringe (**D**) and subsequently passed through a 0.45 µm syringe filter (**E**).

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
