# Peer review of "Stability of African Swine Fever Virus in Soil and Options to Mitigate the Potential Transmission Risk"

_pathogens, 2020, doi:10.3390/pathogens9110977_

Round 1

Reviewer 1 Report

The topic of the article is up to date and has great practical value. Unfortunately, there is still a lot of unknown in terms of african swine fever. In any case, the article would have even greater value if the experiment was carried out in nature where it is exposed to all natural influences.  

Author Response

Reviewer 1

Comment: The topic of the article is up to date and has great practical value. Unfortunately, there is still a lot of unknown in terms of african swine fever. In any case, the article would have even greater value if the experiment was carried out in nature where it is exposed to all natural influences.

Response: We thank Reviewer 1 for the positive assessment of our manuscript! Unfortunately, due to biosafety concerns, we are not able to carry out an experiment like this outside of a containment laboratory.

(see also rebuttal as attachment)

Reviewer 2 Report

This manuscript describes stability of ASFV in different conditions, soils and susceptibility of disinfectants. Study is interesting but there are several issues.  .

  1. Figure 1 describes genome copy and virus titer in blood and yard soil with different temperature. As time goes, viruses will not survive and titer should be decreased as authors shown. But some point like 1 week after virus titer is suddenly increased in yard soil sample. What could be the reason?

It is also well known ASFV is very stable. Reference studies results saying that at 4°C it is also very stable when contained in medium; it remains infectious for at least 61 weeks (1 year and 2 months). In higher temperatures ASFV is inactivated relatively quickly. At 37°C traces of viable virus could be found after 22 days, at 56°C after 1 h, but at 60°C no longer than after 15 min. Authors results shown all viruses mostly dead in 4 weeks. Of course authors samples are different from lab cultured virus. What is your answer?

As the common sense, genome would be higher than virus titer although it is not essential but still easy to guess because genome copies contain both live or dead viruses together.. But authors results shown virus titer is higher than genome copies in several places. How you gonna explain?

  1. Figure 2 describes virus recovery from different types of soils. Trials were OK but what could be the issue on this results in scientifically? Of course wild boar can move everywhere and ASFV can be remain different places. As authors shown in figure 1, ASFV remained up to 1 week in yard soil, but no virus were found in forest soil even at 0 hour? What is the reason?
  2. There is no enough explanation on figure 3 results and materials and methods. Authors saying improved cell culture technique using WSL cells. What is the improved technique?

Author Response

Reviewer 2

Comment: This manuscript describes stability of ASFV in different conditions, soils and susceptibility of disinfectants. Study is interesting but there are several issues.

Figure 1 describes genome copy and virus titer in blood and yard soil with different temperature. As time goes, viruses will not survive and titer should be decreased as authors shown. But some point like 1 week after virus titer is suddenly increased in yard soil sample. What could be the reason?

Response: This is not so much an increase in titer, but rather an increase in variability between replicates, which highlights the challenges of using macrophages to cultivate ASFV from soil matrices. Macrophages do not divide, they are highly sensitive to environmental influences, and most importantly, there is considerable variability among macrophages cultivated from different swine. Unfortunately, we did not obtain enough macrophages from one animal to complete all titrations, introducing this variability in the titers.

The viral genome content stays relatively constant between the samples, but the ability to cultivate ASFV from soil in macrophages is not consistent. This was a first experiment to investigate the experimental possibility to detect live ASFV. Later on, we improved our virus isolation and titration protocol by using WSL cells.

Comment: It is also well known ASFV is very stable. Reference studies results saying that at 4°C it is also very stable when contained in medium; it remains infectious for at least 61 weeks (1 year and 2 months). In higher temperatures ASFV is inactivated relatively quickly. At 37°C traces of viable virus could be found after 22 days, at 56°C after 1 h, but at 60°C no longer than after 15 min. Authors results shown all viruses mostly dead in 4 weeks. Of course, authors samples are different from lab cultured virus. What is your answer?

Response: Sterile buffered culture medium is a much gentler matrix than soil. Temperature is certainly not the only factor at play here. We could clearly see the pH of soil negatively affecting virus stability. Other microorganisms live in the soil and can change that microenvironment and affect virus stability. You can also see that in these experiments, virus titers in pure blood (process controls) remained stable over the three-week storage period at room temperature.

Comment: As the common sense, genome would be higher than virus titer although it is not essential but still easy to guess because genome copies contain both live or dead viruses together. But authors results shown virus titer is higher than genome copies in several places. How you gonna explain?

Response: For every soil sample, one blind passage was always made in macrophages or WSL cells (see section 4.7 in the manuscript). To increase the detection limit, the virus was amplified before it was titrated, therefore titers can be higher than the genome copy numbers measured in the unamplified samples. We have revised the text to make this clearer.

Comment: Figure 2 describes virus recovery from different types of soils. Trials were OK but what could be the issue on this results in scientifically? Of course wild boar can move everywhere and ASFV can be remain different places. As authors shown in figure 1, ASFV remained up to 1 week in yard soil, but no virus were found in forest soil even at 0 hour? What is the reason?

Response: We believe the low pH of forest soil contributed to the decreased stability and survival of ASFV there. This is also mentioned in the discussion (line 193).

Comment: There is not enough explanation on figure 3 results and materials and methods. Authors saying improved cell culture technique using WSL cells. What is the improved technique?

Response: The first advantage of using WSL cells is animal welfare, as we no longer need to take blood from live pigs. Unlike macrophages, the WSL cells divide in culture, so more cells are available, and they are more robust. Virus isolation is more sensitive on a permanent cell line compared to macrophages. The cost and time required are also decreased by using WSL cells. Macrophages take a long time to isolate, harvest, and cultivate, and they need GM-CSF for stimulation, whereas WSL cells just need culture media and serum. We did not include this discussion in the paper because the focus of the paper is ASFV in soil, not virus isolation and titration techniques; however, we have added some more detail to lines 118/119 to indicate the fundamental difference between macrophages and WSL cells.

(see also rebuttal as attachment)

Reviewer 3 Report

The manuscript by Carlson et al. is well written and addresses the survival of African swine fever (ASFV) in different soils. This is particularly important for current ASF outbreaks in Europe, where the disease is mainly affecting wild boar, with carcasses and contaminated soil potentially playing an important role in transmission. The experiments are well designed and the data is well presented, therefore I recommend the manuscript to be accepted.

Author Response

Reviewer 3

Comment: The manuscript by Carlson et al. is well written and addresses the survival of African swine fever (ASFV) in different soils. This is particularly important for current ASF outbreaks in Europe, where the disease is mainly affecting wild boar, with carcasses and contaminated soil potentially playing an important role in transmission. The experiments are well designed and the data is well presented, therefore I recommend the manuscript to be accepted.

Response: We thank Reviewer 3 for the positive assessment of our manuscript!

Reviewer 4 Report

The authors have presented an interesting article dealing with ASF survival in different soil matrices and the effect of disinfection treatment on ASF inactivation. The article brings new and important knowledge important for risk assessment and implementation of adequate mitigation measures.

I have mostly minor comments that are intended to improve clarity of the manuscript with some major comments related to discussing study design limitations.

Materials and methods

Why haven't you repeated a yard soil from pilot experiment since the conditions in the second experiment had changed?

Please explain the rationale behind the experimental conditions (soil weight and infectious blood volume). You have changed the conditions between Experiment 1 and 2.

Please provide details on why have you used the CD2v-deleted ASFV Kenya virus isolate for disinfection treatment experiment and not the Armenia08 virus isolate.

Since paragraphs 4.3, 4.4, 4.5 bring the information on study design and sample preparation without mentioning isolation procedures using macrophages, please rewrite the title of each paragraph

Experiment 2.: What was the media used; was it the same as described in the Experiment 1?

Paragraph 4.6: The title is misleading, please rewrite. In lines 286-287 you mention cells that were not mentioned above. Please explain.

Paragraph 4.7: Titration procedures misses clarity since it is combined within the subparagraph describing virus isolation on WSL cells. There are no details on titration using porcine macrophages. Since I have recommended omitting details on cells used for virus recovery from titles of paragraphs 4.3-4.5, please explain those details in this paragraph (which virus is combined with each cell type).

Paragraph 4.8: Please provide more details on the concentration of primers and probe you have used and thermal conditions, at least an adequate citation. There are some uncertainties regarding the ASF DNA standard generation...you have used the ASF whole genome length for calculation? Since you have extracted a DNA from PBMC culture supernatant, what was the influence of residual DNA of cell origin on measured DNA concentration values? Why haven't you opted for plasmid based DNA standards?

Results

Lines 63-64. You are mentioning four week timeframe. This information is missing in the corresponding paragraph in materials and methods. Furthermore, you are mentioning blood-only control in the Experiment 1., however that is not clear from the information provided in the materials and methods.

Description of Figure 1. in lines 82-83 is not clear enough. Please rewrite the sentence.

Lines 116-117. You are mentioning yard soil which is not described as included in experiment 4. in the materials and methods. Please clarify

Line 131. You are mentioning potting soil only later in this paragraph, no information was provided at the beginning (lines 116-117).

You are starting the paragraph 2.3 with the sentence that is indicating different objective of this experiment.  The objective from my point of view is the evaluation of disinfection treatments on ASF inactivation in different soil matrices. Please clarify.

Overall, the Figure 4. is not clear enough in visualizing disinfection treatments. Use different symbols instead of coloured circles with grouping instead of overlapping. Moreover you have included an open circle for each replicate which should be removed as you already depicted error bars. Please clarify the genome detection in the part of experiment with treated samples (Fig.4 A)...it is not clear whether you have detected genome throughout the experiment as indicated in the text or just after 1 and 3 hours? Why the blood-only control that was untreated was not tested after 1 and 3 h (Fig. 4 A and B)?  Letter “a)” and “b)” should be removed from the Figure 4.

Discussion

Line 170. Please provide the full name for HAT abbreviation. No information is provided on HAT in the materials and methods.

Line 190. You are using the term epizootic in contrast to the term epidemic you have used in the line 38. Please use either of two and provide and explanation.

Please discuss the possible limitations of a study design. The ratio between soil weight and infectious blood volume used in different experiments in contrast to the corresponding presumable ratio in natural conditions. A depth of carcass fluids drainage in different soils and possible impact on disinfection effectiveness. Furthermore please explain possible limitation of tests used to detect infectious virus compared to bioassay.

Author Response

Reviewer 4

Comment: The authors have presented an interesting article dealing with ASF survival in different soil matrices and the effect of disinfection treatment on ASF inactivation. The article brings new and important knowledge important for risk assessment and implementation of adequate mitigation measures.

Response: We thank the reviewer for the positive assessment of our manuscript!

Comment: I have mostly minor comments that are intended to improve clarity of the manuscript with some major comments related to discussing study design limitations.

Materials and methods

Comment: Why haven't you repeated a yard soil from pilot experiment since the conditions in the second experiment had changed?

Response: The time between when the first experiment was completed and the next was more than 6 months. Since the soil changes with the seasons, we would not have had the same sample as before. There is no way to truly preserve the soil we originally sampled; it sat in a bag and grew mold. Therefore, we introduced the potting soil, hoping it would resemble yard or garden soil with a neutral pH, while also being a more controlled type of sample that we could go back to if needed.

Comment: Please explain the rationale behind the experimental conditions (soil weight and infectious blood volume). You have changed the conditions between Experiment 1 and 2.

Response: We found that some of the forest soils were so dry that blood and media were completely absorbed (see line 283). We had to increase the volumes so that when the soil was centrifuged there was enough liquid released to filter and use for virus isolation and DNA extraction.

Comment: Please provide details on why have you used the CD2v-deleted ASFV Kenya virus isolate for disinfection treatment experiment and not the Armenia08 virus isolate.

Response: Recombinant viruses that express a fluorescent reporter gene in infected cells are very useful for studies that include many virus isolations and titrations, because the readout is much quicker and requires no additional treatment of the cultures. In addition, this virus grows very well and to high titers in WSL cells, which are much easier to handle than primary macrophages. When we conducted the first experiments, however, this virus was not available to us, otherwise we would have used it for the entire study. It would have been even better to have a recombinant Armenia reporter virus, but that was not available at the time either.

Comment: Since paragraphs 4.3, 4.4, 4.5 bring the information on study design and sample preparation without mentioning isolation procedures using macrophages, please rewrite the title of each paragraph

Response: The macrophage cultivation is described in 4.7. We believe it is critical to mention macrophages in the title for each experiment, however, so that readers can quickly identify which experiment was done in macrophages and which was not.

Comment: Experiment 2.: What was the media used; was it the same as described in the Experiment 1?

Response: All macrophage cultivation and virus isolation was done the same way for experiments 1 to 3; it is described in section 4.7. We edited the text to make this clearer.

Comment: Paragraph 4.6: The title is misleading, please rewrite. In lines 286-287 you mention cells that were not mentioned above. Please explain.

Response: The title has been changed. All cell culture is described in section 4.7.

Comment: Paragraph 4.7: Titration procedures misses clarity since it is combined within the subparagraph describing virus isolation on WSL cells. There are no details on titration using porcine macrophages. Since I have recommended omitting details on cells used for virus recovery from titles of paragraphs 4.3-4.5, please explain those details in this paragraph (which virus is combined with each cell type).

Response: The section has been divided and expanded as requested by the reviewer.

Comment: Paragraph 4.8: Please provide more details on the concentration of primers and probe you have used and thermal conditions, at least an adequate citation.

The requested information was added to the text.

Comment: There are some uncertainties regarding the ASF DNA standard generation...you have used the ASF whole genome length for calculation? Since you have extracted a DNA from PBMC culture supernatant, what was the influence of residual DNA of cell origin on measured DNA concentration values? Why haven't you opted for plasmid based DNA standards?

Response: We have added some detail to the description to address the reviewer’s questions. The reviewer is correct in that we did not account for residual DNA of cellular origin. However, we considered this acceptable because we only use the standard for harmonization between PCR runs, not for absolute quantification.

Results

Comment: Lines 63-64. You are mentioning four week timeframe. This information is missing in the corresponding paragraph in materials and methods. Furthermore, you are mentioning blood-only control in the Experiment 1., however that is not clear from the information provided in the materials and methods.

Description of Figure 1. in lines 82-83 is not clear enough. Please rewrite the sentence.

Response: We appreciate the reviewer’s attention to detail! These oversights have been corrected in the revised manuscript and we have revised the figure caption.

Comment: Lines 116-117. You are mentioning yard soil which is not described as included in experiment 4. in the materials and methods. Please clarify

Line 131. You are mentioning potting soil only later in this paragraph, no information was provided at the beginning (lines 116-117).

Response: Yes, it should say “commercial potting soil” and not “yard soil” at the beginning of the paragraph, too. That has been corrected.

Comment: You are starting the paragraph 2.3 with the sentence that is indicating different objective of this experiment. The objective from my point of view is the evaluation of disinfection treatments on ASF inactivation in different soil matrices. Please clarify.

Response: It is really both. The headings and introductory text in the methods and results sections have been changed to better reflect that.

Comment: Overall, the Figure 4. is not clear enough in visualizing disinfection treatments. Use different symbols instead of coloured circles with grouping instead of overlapping. Moreover you have included an open circle for each replicate which should be removed as you already depicted error bars. Please clarify the genome detection in the part of experiment with treated samples (Fig.4 A)...it is not clear whether you have detected genome throughout the experiment as indicated in the text or just after 1 and 3 hours? Why the blood-only control that was untreated was not tested after 1 and 3 h (Fig. 4 A and B)? Letter “a)” and “b)” should be removed from the Figure 4.

The figure has been revised as suggested. We now use different symbols with horizontal spacing between groups. The reviewer is correct in that having both error bars and replicates is redundant, so we opted to remove the error bars. The blood-only control was not tested during the first 24 hours because the virus is highly stable in blood and no drop in titer was expected this early, which is also borne out by the observed stability over the following weeks.

Discussion

Comment: Line 170. Please provide the full name for HAT abbreviation. No information is provided on HAT in the materials and methods.

Response: The requested information has been added to section 4.7.

Comment: Line 190. You are using the term epizootic in contrast to the term epidemic you have used in the line 38. Please use either of two and provide and explanation.

Response: This has been changed to epizootic throughout.

Comment: Please discuss the possible limitations of a study design. The ratio between soil weight and infectious blood volume used in different experiments in contrast to the corresponding presumable ratio in natural conditions. A depth of carcass fluids drainage in different soils and possible impact on disinfection effectiveness. Furthermore please explain possible limitation of tests used to detect infectious virus compared to bioassay.

Response: We have included these considerations in the revised discussion (lines 180-181, 193-195, 223/224).

(see also full rebuttal as attachment)

Round 2

Reviewer 2 Report

good information